# Urine metabolomics of rats with chronic atrophic gastritis

**Guo-Xiu Zu** [1], **Qian-Qian Sun** [1], **Jian Chen** [1,2], **Xi-Jian Liu** [1], **Ke-Yun Sun** [1], **Liang-Kun Zhang** [1], **Ling Li** [1], **Tao Han** [3]*, **Hai-Liang Huang** [4]*

**1** Department of Traditional Chinese Medicine, Shandong University of Traditional Chinese Medicine, Jinan, Shandong, China, **2** Affiliated Central Hospital of Shandong First Medical University, Shandong First Medical University, Jinan, Shandong, China, **3** Graduate Office, Shandong University of Traditional Chinese Medicine, Jinan, Shandong, China, **4** Department of Rehabilitation Medicine, Shandong University of Traditional Chinese Medicine, Jinan, Shandong, China

☯ These authors contributed equally to this work.
* 60012002@sdutcm.edu.cn (TH); 06000031@sdutcm.edu.cn (HLH)

## Abstract

### Background/aim

To use liquid chromatography-mass spectrometry (LC-MS) to identify endogenous differential metabolites in the urine of rats with chronic atrophic gastritis (CAG).

### Materials and methods

Methylnitronitrosoguanidine (MNNG) was used to produce a CAG model in Wistar rats, and HE staining was used to determine the pathological model. LC-MS was used to detect the differential metabolic profiles in rat urine. Diversified analysis was performed by the statistical method.

### Results

Compared with the control group, the model group had 68 differential metabolites, 25 that were upregulated and 43 that were downregulated. The main metabolic pathways were D-glutamine and D-glutamic acid metabolism, histidine metabolism and purine metabolism.

### Conclusion

By searching for differential metabolites and metabolic pathways in the urine of CAG rats, this study provides effective experimental data for the pathogenesis and clinical diagnosis of CAG.

## Introduction

Chronic atrophic gastritis (CAG) is a type of atrophy of gastric mucosal epithelial cells and glands where the number of glands is reduced, the mucosal layer thins, and the mucosal

**Data Availability Statement:** All relevant data are within the paper and its Supporting information files.

**Funding:** The grant of the Preliminary Mechanism and Efficacy Evaluation by the excellent scientific

research and innovation teams at Shandong University of Traditional Chinese Medicine in the treatment of major diseases No. 220316 and Shandong province key research and development plan 2016CYJS08A01-6"Establishment of Clinical Diagnosis and Treatment Standard System of Precise Prescription for the Prevention and Treatment of Gastric Precancerous Lesions by Jingfang Pinellia Xiexin Decoction.

**Competing interests:** The authors have declared that no competing interests exist.

muscle layer thickens and may be accompanied by intestinal metaplasia and dysplasia. Digestive system diseases [1] mainly have the clinical manifestations of bloating, fullness of the stomach, belching, pain in the upper abdomen, loss of appetite, weight loss, etc. CAG has a wide variety of factors and is a common and frequently occurring clinical disease with a 2.55%- 7.46% canceration rate [2]. In 1978, the World Health Organization officially defined chronic atrophic gastritis as a precancerous state. The active treatment of CAG in clinical practice is an important node to block its development into gastric cancer.

As an important branch of systems biology, metabolomics technology is unique because it does not require the establishment of a large database of expressed gene sequences [3]. Metabolomics can express the physiological and biochemical state of the body through biological metabolic structure to better analyze pathogenesis. Among its advantages, liquid chromatography-mass spectrometry (LC-MS) technology can be directly used to analyze biological metabolites to obtain final analysis results with the advantage of finding subtle changes in gene and protein expression during biological metabolism. Thus, LC-MS has become the most commonly used analytical technique in metabolomics research [4]. This study explains the molecular mechanism of action and metabolic pathways of chronic atrophic gastritis through pharmacodynamics and LC-MS.

## Materials and methods

### Animals

Twenty SPF grade Wistar male rats, 6 weeks old, $180 \pm 20$ g, were provided by Shandong Pengyue Experimental Animal Co., Ltd. [SCXK (Lu) 20140007]. The feeding environment was a temperature of $26°C \pm 2°C$, humidity $50 \pm 10\%$, and light illumination/dark cycle 12 h. The experiment started after 7 d of adaptive feeding from the time of purchase. During the period, the animals has free access to food and drinking water, and the experiment met animal ethical requirements.

### Experimental reagents and instruments

Methylnitronitrosoguanidine (manufactured by Tokyo Chemical Industry Co., Ltd., NH8JH-DR), vetzyme tablets (Lepu Hengjiuyuan Pharmaceutical Co., Ltd., 20170401), ranitidine hydrochloride capsules (Tianjin Pacific Pharmaceutical Co., Ltd., 20170601), and ammonium hydroxide (Shanghai Wokai Biotechnology Co., Ltd., 20170220) were used. Anhydrous ethanol (Tianjin Fuyu Fine Chemical Co., Ltd., 20170808), methanol (Woke), acetonitrile and formic acid (Aladdin), ammonium formate (Sigma), hematoxylin staining solution, eosin staining solution, differentiation solution, blue back solution (Hebei Bohai Biological Engineering Development Co., Ltd.), and xylene (Tianjin Yongda Chemical Reagent Co., Ltd.) were also utilized in this study.

A refrigerated centrifuge (Eppendorf, H1650-W), mixer (Vortex Mixer, QL-866), liquid chromatograph (Thermo, UltiMate 3000) and mass spectrometer Thermo (Q Exactive Focus) were instruments used in this study.

### Animal model

Twenty Wistar rats were prepared, and 10 rats were randomly selected as a blank group. Normal diet was fed until the materials were collected. The remaining rats were model rats according to the following method [5]: rats were given 120 μg/mL MNNG from the 1st day of modeling, given 0.1% ammonia water freely for 24 h fed with 0.03% ranitidine feed using the hunger and satiety method (full food for 2 d, fasting for 1 d), an each given 2 ml of 40% ethanol

on the fasting day. The above operation lasted for 16 weeks and each rat was weighed twice a week during the modeling process. During the experiment, the weight, coat color and behavior of the rats were observed.

Example ethics statement: This study was carried out in strict accordance with the recommendations in the Guidelines for ethical review of experimental Animal welfare (National Standard: GB/T 35892–2018). The protocol was approved by the Ethics Review Center of Shandong University of Traditional Chinese Medicine (Protocol Number: SDUTCM20190402003). All surgery was performed under sodium pentobarbital anesthesia, and all efforts were made to minimize suffering.

## Urine collection and preparation

Before sampling, they fasted for 24 hours, drank normal water, collected urine, followed by anaesthesia with 2% pentobarbital sodium, blood collection of spleen, stomach and liver, and subsequent death. Urine was centrifuged at 2500 rpm at room temperature for 1 hour in the morning, and the supernatant was divided into centrifuge tubes; each tube was > 0.3 ml. The urine samples were melted at 4˚C and 100 μL of each sample was placed into a 1.5 mL centrifuge tube, 100 μL of ddH$_2$O was added followed by shaking for 5 min to fully absorb and centrifugation at 10000 g and 4˚C for 10 min. Then, a 0.22 μm membrane was used to filter the supernatant to obtain the samples to be tested; 20 μL of the synthetic QC samples were extracted from each sample to be tested, and the remaining samples were tested by LC-MS.

## LC-MS chromatographic mass spectrometry conditions

A Thermo Ultimate 3000 chromatograph and an ACQUITY UPLC® HSS T3 1.8 μm (2.1 × 150 mm) chromatographic column were used with an autosampler temperature of 8˚C, a flow rate of 0.25 mL/min, and a column temperature of 40˚C. The sample was eluted with an injection volume of 2 μl, and the positive mode mobile phases were 0.1% formic acid in water (A) and 0.1% formic acid in acetonitrile (B). The gradient elution program was 0 ~ 2 min, 2% B; 2 ~ 10 min, 2% ~ 50% B; 10 ~ 15 min, 50% ~ 98% B; 15 ~ 20 min, 98% B; 20 ~ 22 min, 98% ~ 2% B; 22 ~ 25 min, 2% B. The negative mode mobile phases were 5 mM ammonium formate (A) and acetonitrile (B). The gradient elution program was 0 ~ 2 min, 2% B; 2 ~ 10 min, 2% ~ 50% B; 10 ~ 15 min, 50% ~ 98% B; 15 ~ 20 min, 98% B; 20 ~ 22 min, 98% ~ 2% B; 22 ~ 25 min, 2% B [6]. The Thermo Q Exactive Focus mass spectrometer was operated with the following conditions: electrospray ion (ESI) source, positive and negative ion ionization mode, positive ion spray voltage of 3.50 kV, negative ion spray voltage of -2.50 kV, sheath gas of 30 arb, auxiliary gas of 10 arb, capillary temperature of 325˚C, full scan with a resolution of 70,000, scan range of m/z 81–1000, secondary cracking with HCD, collision voltage of 30 eV, and dynamic exclusion to remove unnecessary MS/MS information.

## Data processing

The analysis software used for multidimensional statistical analysis is SIMCA-P (V13.0). In addition, the calculation of P value is t-test. The t-test that we use for univariate statistical analysis, in general, t-test is $p < 0.05$ is significant, $p < 0.01$ is very significant, biological statistical methods are basically such a display of differences; This section provides references for significance of biological repeated screening p values. The obtained raw data was converted to mzXML format with ProteoWizard software (v3.0.8789) [7] and the RCMS (v3.3.2) XCMS package was used for peak identification, peak filtering, and peak alignment analysis. The main parameters were bw = 5, ppm = 15, peakwidth = c (10, 20), mzwid = 0.015, mzdiff = 0.01, and method = centWave, which includes the mass to charge ratio (m/z) and information data

matrix such as retention time (rt) and intensity. According to the results, the original peak area, which is the relative strength value, is calculated, then the original peak area is standardized, batch normalization of peak area of data. Data analysis is based on standardized data. Multidimensional statistical analysis was performed based on the data after standardized processing.

LC-MS data is carried out on the basis of normalization, eliminating very few data that do not exist or have too low strength. The experiment adopts most data retained after QC, QA and normalization processing. Prior to urine metabolomics analysis, the Proteowizard software (V3.0.8789) was used to convert the obtained original data into mzXML format (XCMS input file format). Using R (v3.3.2) XCMS package is used to identify the peaks identification, peaks filtration, peaks alignment, the main parameters are bw = 5, PPM = 15, peakwidth = c (10, 20), mzwid = 0.015, mzdiff = 0.01, the method = centWave. The data matrix, including mass to charge ratio (M/Z), retention time (RT) and intensity, is obtained. In the positive ion mode 22,540 precursor molecules and the negative ion mode 18,837 precursor molecules were obtained. The data were exported to Excel for subsequent analysis. In order to make comparison of data of different orders, batch normalization of data regarding peak area was conducted.

## Results

### General situation

Control group: In good condition, sturdy body, strong limbs, neat, supple and shiny fur, mental state is excellent, responsive to external conditions, body weight gradually increases, and the stool is normal. Model group: Poor condition, thin body, weak limbs, messy fur, dryness, and dullness, poor mental state, drowsiness, unresponsive to external conditions, insignificant changes in body mass, slower rise, and less stool that is hard. Body mass changes are shown in Fig 1. In the registration of weight changes at 16 weeks, normality test was performed first, and then one-way ANOVA was used to conduct statistical test of weight between the blank group and the model group, and $^*P<0.01$ was found between the two groups, the difference was statistically significant.

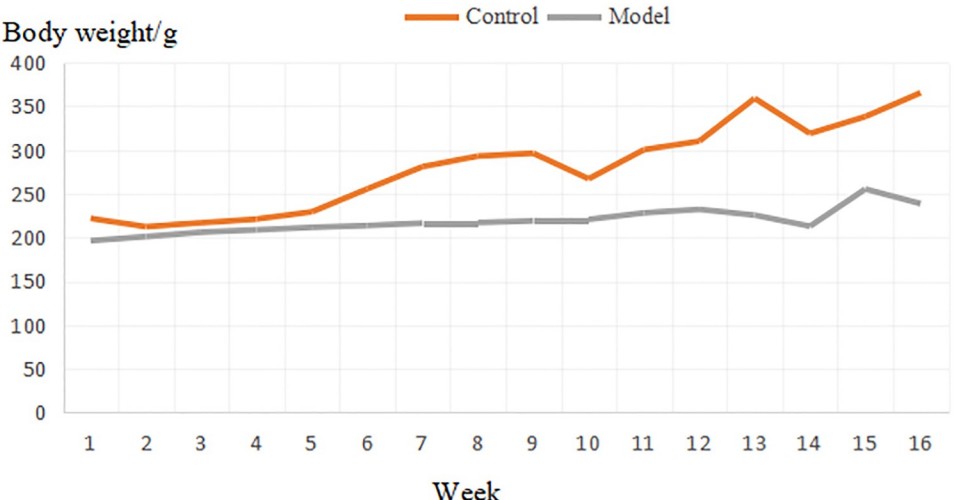

**Fig 1. Mass variation diagram of the control group and model group.**

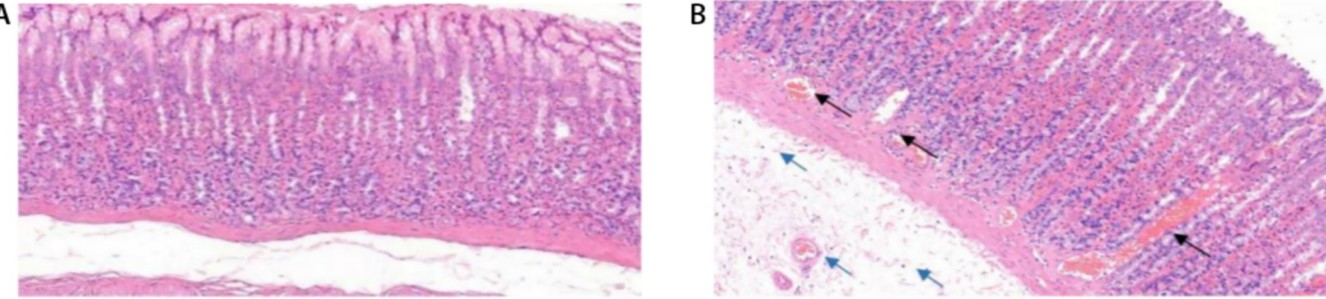

**Fig 2. HE staining pathological sections.** A. HE Staining Pathological Sections of Gastric Mucosa in the Control Group (×200); 2-B: HE Staining Pathological Sections of Gastric Mucosa in the Model Group (×200).

## Observation of pathological tissues

As shown in Fig 2A, the gastric tissue mucosa lamina propria in the blank group pathological section is rich in gastric glands that are closely arranged with a normal structure, and the gastric gland epithelial cells have a normal morphology. In the model group, the lamina propria were loosely arranged, the lamina propria of the gastric mucosa was severely congested (black arrow), and there were a large number of inflammatory cells (blue arrow) under the mucosa with edema, as shown in Fig 2B.

## Chromatogram in total ion mode

The components separated by chromatography entered into mass spectrometry (MS) analysis, and data collection was performed by continuous scanning of the mass spectrum. The intensity is on the ordinate, and the time is on the abscissa. The resulting spectrum is the base peak chromatogram (BPC); see Fig 3A and 3B (G: model group, H: control group).

## Urine metabolomics analysis in positive ion mode

After data pretreatment (format conversion peak recognition, filtering alignment and normalization), the data screened out have strong repeatability and good effect for urine metabolomics analysis After the data were preprocessed, the principal component analysis (PCA) method was used to explore CAG urine in positive ion mode. Changes in the fluid metabolism profile yielded a model with three principal components ($R^2 = 0.548$) and a score chart reflecting the degree of dispersion between groups, as shown in Fig 4A. The PCA score graph shows that most samples are within the ellipse of the 95% confidence interval except for individual outliers. The PCA score graph shows that the urine samples of the two groups are significantly separated and are statistically significant. Furthermore, PLS-DA and OPLS-DA analysis methods (Fig 4B and 4C) were used to remove information that was not related to sample classification, and pattern discrimination analysis was performed on the full spectrum of the urine. Permutation is the result of 200 permutation tests, PLD-DA permutation test in positive ion mode is $R^2 = (0.0, 0.91)$, $Q^2 = (0.0, 0)$ PLD-DA permutation test in negative ion mode is $R^2 = (0.0, 0.9)$, $Q^2 = (0.0, -0.39)$ The results showed that the two groups of samples could be significantly separated. In order to check whether the repeatability of the model is good and ensure the reliability of the data model, a permutation test was performed on the model (Fig 4D). The above results show that the multivariate data model of urine samples meets the parameter standard, indicating that the model has high stability and good predictive ability.

**Base Peak Chromatogram**

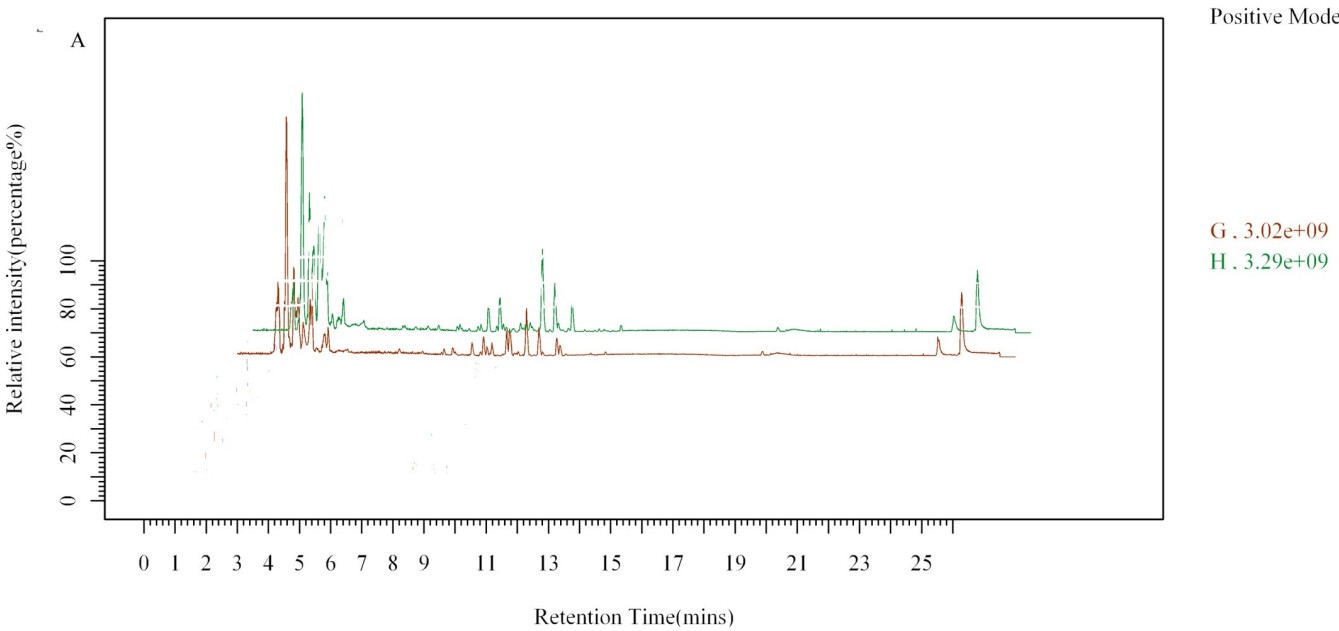

**Base Peak Chromatogram**

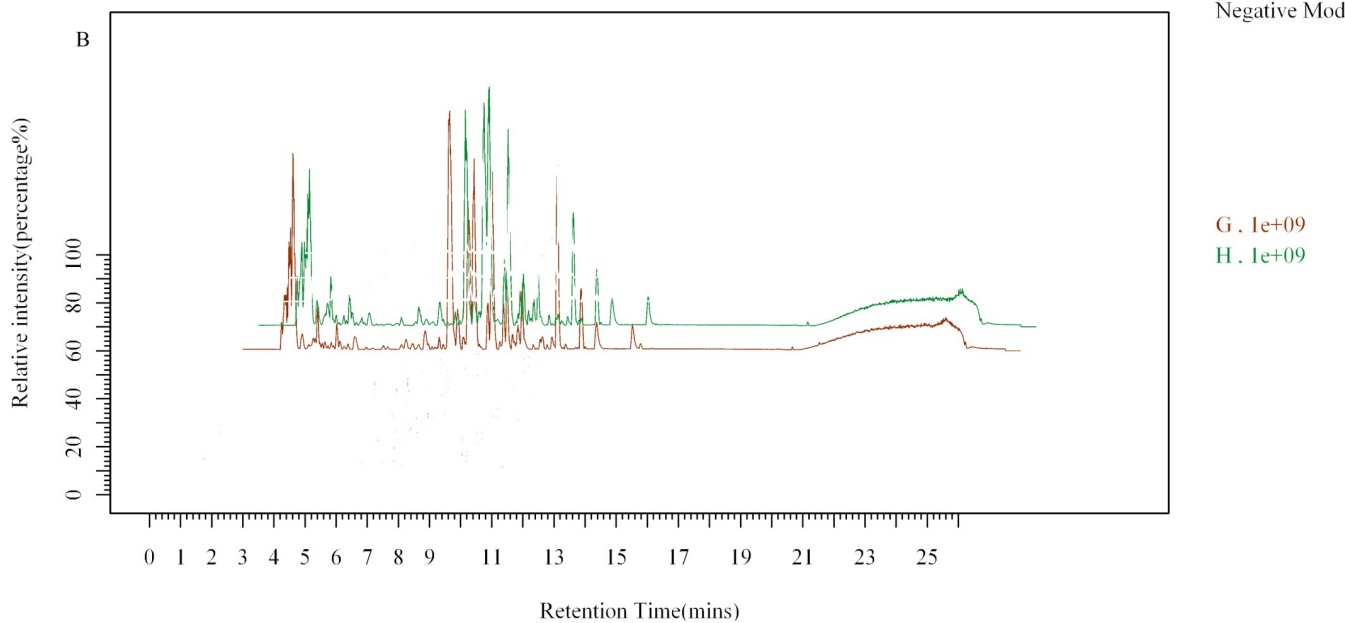

**Fig 3. Chromatogram in total ion mode.** 3-A: Typical Sample BPC in Positive Ion Mode, 3-B: Typical Sample BPC in Negative Ion Mode.

## Analysis of urine metabolomics in negative ion mode

The PCA method was used to explore changes in the CAG urine metabolic spectrum. After data preprocessing, a model with 3 principal components ($R2 = 0.515$) and the degree of dispersion between groups were obtained from the score chart. The PCA score graph shows that

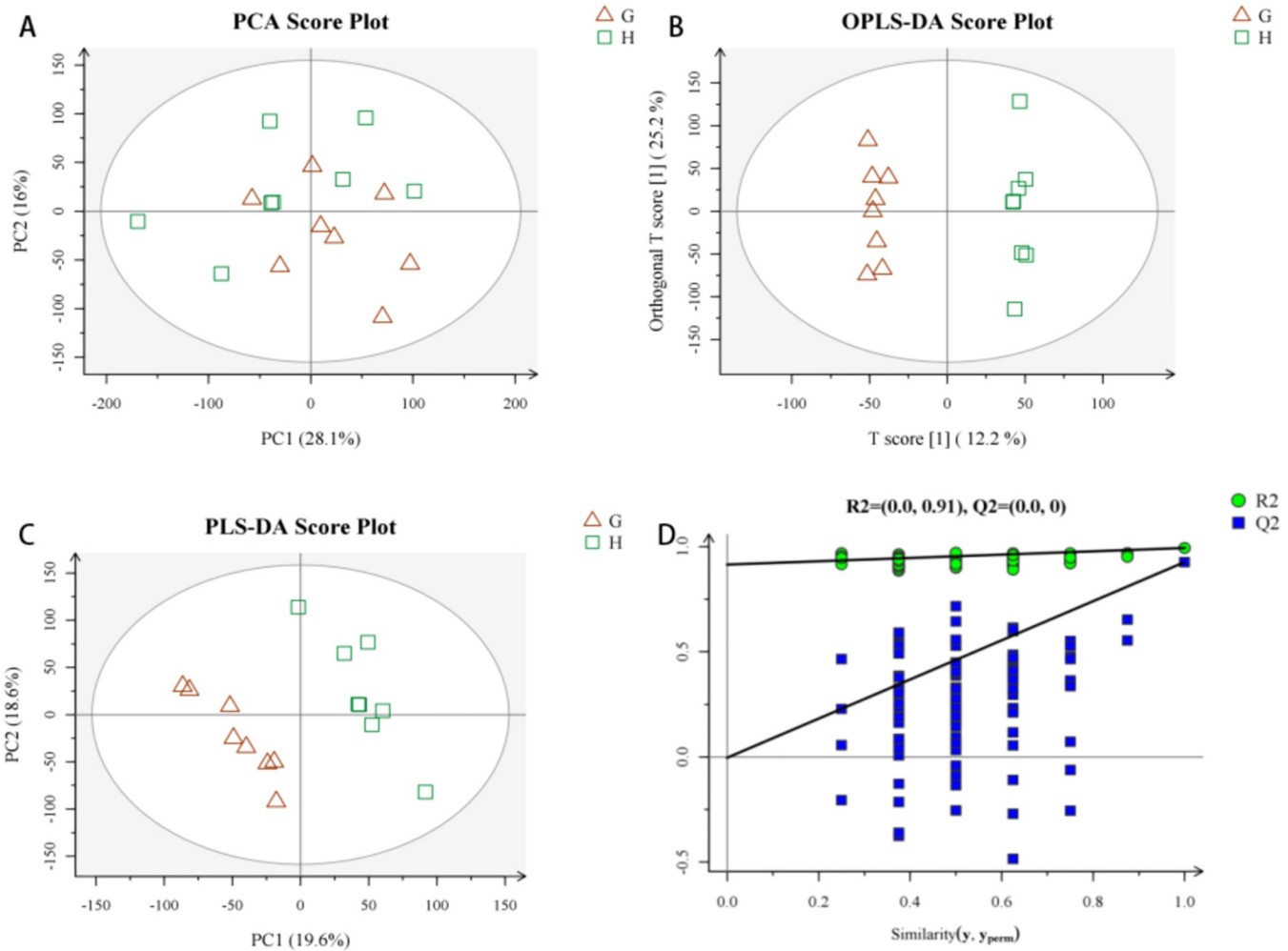

**Fig 4. Urine metabolism profile of CAG model rats in positive ion mode.** 4-A: PCA Scores, 4-B: PLS-DA Scores, 4-C: OPLS-DA Scores, 4-D: Replacement Test of the CAG Model Urine Fit Model in Positive Ion Mode.

most samples fall within the ellipse of the 95% confidence interval, with only a few outliers. The PCA score (Fig 5A) graph shows the spatial distribution of the urine samples of the two groups, which can be significantly separated. PLS-DA and OPLS-DA analysis methods were used to further analyze the full spectrum of urine, and the results showed that the two groups of samples could be significantly separated (Fig 5B and 5C). In order to test whether the repeatability of the model is good and to ensure reliability of the data model, the model was replaced and verified (Fig 5D). The intercept of Q2 is negative, indicating that the model is valid. The above results indicate that the multivariate data model of urine samples meets the parameter standard, indicating that the model has high stability and good predictive ability.

## Extraction and analysis of differential metabolites

From the PCA, PLS-DA, OPLS-DA analysis model group and blank group, the screening conditions were in accordance with a P-value $\leq 0.05$, VIP $\geq 1$ [6], and molecular weight error $< 20$ ppm). According to the fragmentation information obtained from MS/MS mode, further matching annotations were obtained in the HMDB, METLIN, MassBank, LipidMaps, and

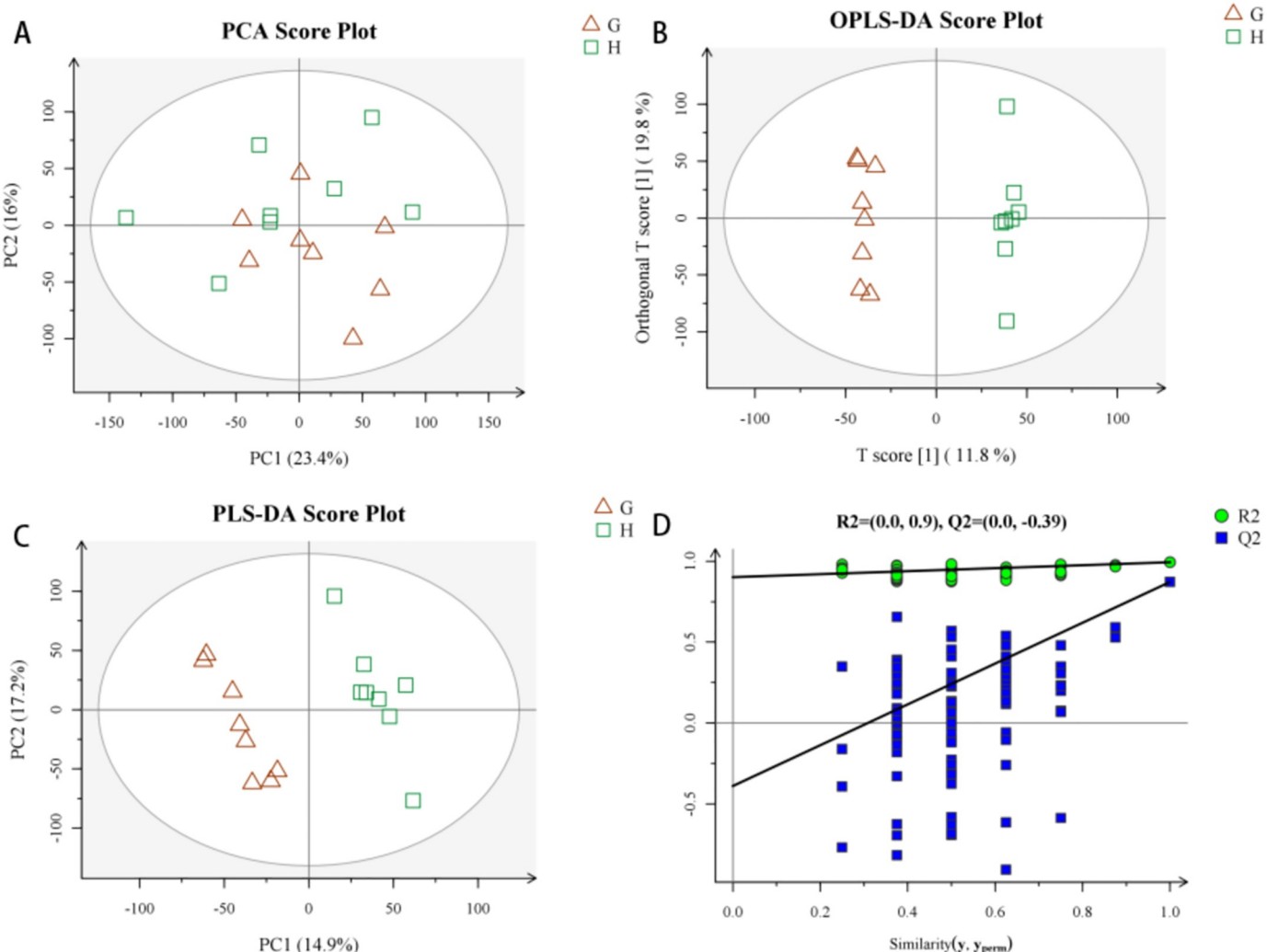

**Fig 5. Urine metabolism profile of CAG model rats in negative ion mode.** 5-A: PCA Scores, 5-B: PLS-DA Scores, 5-C: OPLS-DA Scores, 5-D: Replacement Test of the CAG Model Urine Fit Model in Negative Ion Mode.

mzCloud databases to obtain accurate metabolite information. A total of 68 differential metabolites were screened, of which 25 were upregulated and 43 that were downregulated, compared with metabolites with the same or similar metabolic modes clustered to obtain differential metabolite heat maps and metabolite correlation heat maps (Fig 6). These differential metabolites relied on the Marker-view, KEGG, HMDB, MetaboAnalyst and other databases, which were searched and identified, and the results are shown in Table 1.

## CAG model group urine differential metabolite pathway information

This study mapped the differential metabolites to the KEGG database. There are 23 common metabolic pathways involved in the obtained differential metabolites, as shown in Fig 7: D-glutamine and D-glutamine histidine metabolism, histidine metabolism, purine metabolism, nitrogen metabolism, tyrosine metabolism, arginine-proline metabolism, butyric acid metabolism, biotin metabolism, alanine-aspartic acid-glutamic acid metabolism, ascorbic acid-bitter almond metabolism, niacin-nicotinamide metabolism, pentose-glucuronate interconversion,

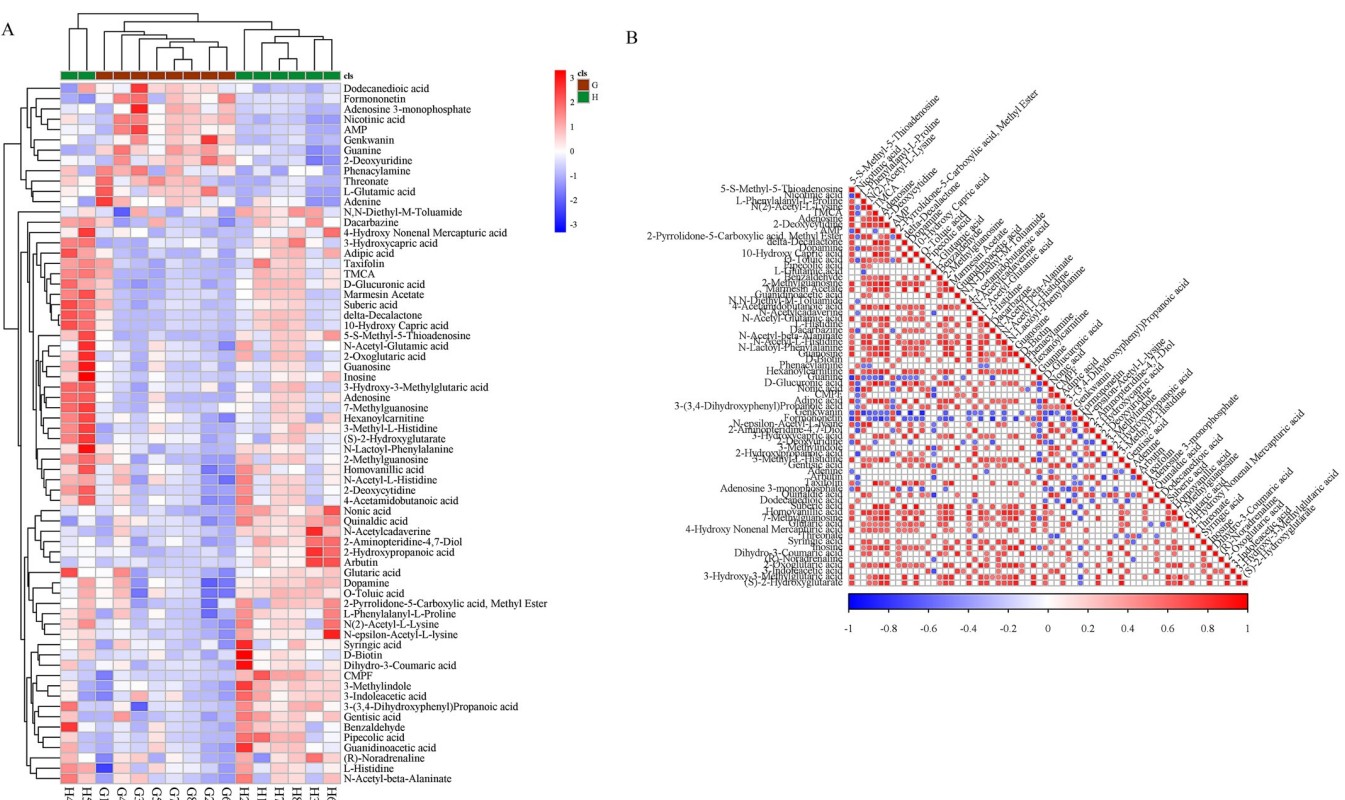

**Fig 6. Heat map of the differential metabolites.** A: Heat Map of the Differential Metabolites in CAG Rats; 6-B: Correlation Heat Map of Differential Metabolites in CAG Rats.

pyrimidine metabolism, lysine degradation, citric acid cycle (TCA cycle), starch-sucrose metabolism, Inositol phosphate metabolism, glutathione metabolism, porphyrin and chlorophyll metabolism, cysteine-methionine metabolism, glycine-serine-threonine metabolism, aminoacyl-tRNA biosynthesis, and tryptophan metabolism. Among them, the metabolic pathways with * P <0.5 and Impact> 0 include D-glutamine and D-glutamic acid metabolism, histidine metabolism, and purine metabolism, as shown in Table 2. In these pathways, D-glutamine and D-glutamic acid metabolism were up-regulated, three metabolites in histamine metabolism were down-regulated, and six metabolites in purine metabolism were down-regulated.

## Discussion

In recent years, metabonomics has been widely used in modern Chinese medicine treatment of chronic gastritis. Cui Jiajia Et al. [8] found that 3 plasma biomarkers (arginine, succinate and 3-hydroxybutyrate) and 2 urine biomarkers (α-ketoglutarate and valine) might be markers of CAG in the study on plasma and urine metabolites. Chen jiaolong [9] by using nuclear magnetic metabonomics technology including observation of the stomach meridian the CAG treatment, use of electroacupuncture in the rat stomach meridians beam door and foot three mile, found that serum ghrelin level stomach metabolism of liver kidney brain cortex spectral change, the rat gastric mucosa arrangement and the thickness of the gastric mucosa has different degrees of improvement, ghrelin and substance P expression in serum increased to normal level, serum glucose glycogen content increased, the stomach tissue of glutathione and

**Table 1. Differential metabolic markers in urine of CAG rats (upregulated ↑, downregulated ↓).**

| chemical compound | m/z | rt | exact mass | chemical formula | Model vs Control_VIP | log2 (FC) | p value |
|---|---|---|---|---|---|---|---|
| (S)-2-Hydroxyglutarate | 147.0287633 | 82.9724 | 148.11402 | C5H8O5 | 1.464880663 | 0.5013 | 0.047581863↓ |
| ®-Noradrenaline | 184.0967985 | 330.939 | 183.205 | C8H11NO3 | 1.526685758 | 0.3797 | 0.037595251↓ |
| 10-Hydroxy capric acid | 189.1483988 | 609.2975 | 188.264 | C10H20O3 | 1.823597606 | 1.5684 | 0.008306808↑ |
| 2-Aminopteridine-4,7-Diol | 178.0362164 | 219.251 | 179.1364 | C6H5N5O2 | 2.016289866 | 1.1144 | 0.003043425↑ |
| 2-Deoxycytidine | 228.0976635 | 149.877 | 227.2172 | C9H13N3O4 | 1.915004809 | 1.3307 | 0.004840515↑ |
| 2-Deoxyuridine | 227.0667048 | 283.209 | 228.202 | C9H12N2O5 | 1.943477192 | -0.979 | 0.004861136↓ |
| 2-Hydroxypropanoic acid | 89.02311403 | 93.0333 | 90.0779 | C3H6O3 | 1.874058772 | 0.3997 | 0.00731866↓ |
| 2-Methylguanosine | 298.1143794 | 220.051 | 297.2675 | C11H15N5O5 | 1.751775025 | 1.367 | 0.012215874↑ |
| 2-Oxoglutaric acid | 145.0131155 | 83.6063 | 146.0981 | C5H6O5 | 1.495118055 | 0.9552 | 0.042475723↓ |
| 2-Pyrrolidone-5-Carboxylic acid, Methyl Ester | 144.0654693 | 163.765 | 143.0582432 | C6H9NO3 | 1.868632508 | 0.4193 | 0.006413493↓ |
| 3-(3,4-Dihydroxyphenyl)Propanoic acid | 181.0496532 | 227.9005 | 182.1733 | C9H10O4 | 2.070984986 | 0.8642 | 0.002077192↓ |
| 3-Hydroxy-3-methylglutaric acid | 161.044468 | 84.9492 | 162.1406 | C6H10O5 | 1.480741362 | 0.7542 | 0.044849483↓ |
| 3-Hydroxycapric acid | 187.1330551 | 667.4145 | 188.264 | C10H20O3 | 1.944065438 | 0.3836 | 0.004843594↓ |
| 3-Indoleacetic acid | 174.0550606 | 463.637 | 175.184 | C10H9NO2 | 1.490647745 | 0.5842 | 0.043203485↓ |
| 3-Methylindole | 130.0650034 | 464.003 | 131.1745 | C9H9N | 1.876426797 | 0.9857 | 0.007221167↓ |
| 3-Methyl-L-histidine | 168.0767831 | 105.457 | 169.18126 | C7H11N3O2 | 1.872481394 | 1.4212 | 0.00738418↑ |
| 4-Acetamidobutanoic acid | 146.0811334 | 296.814 | 145.15648 | C6H11NO3 | 1.641325837 | 0.526 | 0.02089112↓ |
| 4-Hydroxy nonenal Mercapturic acid | 318.1376294 | 559.432 | 319.1453439 | C14H25NO5S | 1.548812817 | 1.0965 | 0.034435408↑ |
| 5-S-Methyl-5-thioadenosine | 298.0965219 | 450.365 | 297.3347 | C11H15N5O3S | 2.284849647 | 1.8528 | 0.000234115↑ |
| 7-Methylguanosine | 296.0994864 | 384.98 | 297.1073186 | C11H16N5O5 | 1.568140916 | 1.549 | 0.031843615↑ |
| Adenine | 134.0460176 | 301.647 | 135.1269 | C5H5N5 | 1.792987662 | -0.7032 | 0.011340028↓ |
| Adenosine | 268.1038713 | 289.983 | 267.24152 | C10H13N5O4 | 1.934286006 | 2.0713 | 0.004285334↑ |
| Adenosine 3-monophosphate | 346.0554562 | 184.5985 | 347.22142 | C10H14N5O7P | 1.68169368 | -1.3227 | 0.019481197↓ |
| Adipic acid | 145.049496 | 93.721 | 146.1412 | C6H10O4 | 2.113128751 | 1.3866 | 0.001516902↑ |
| AMP | 348.0700631 | 175.3335 | 347.2212 | C10H14N5O7P | 1.900260783 | -1.5778 | 0.005302741↓ |
| Arbutin | 271.0820178 | 255.116 | 272.2512 | C12H16O7 | 1.771574784 | 2.3461 | 0.012648447↑ |
| Benzaldehyde | 107.0492548 | 441.291 | 106.1219 | C7H6O | 1.762739968 | 1.0398 | 0.011540491↑ |
| CMPF | 239.0910035 | 181.3095 | 240.2524 | C12H16O5 | 2.124552651 | 1.2416 | 0.001388456↑ |
| Dacarbazine | 183.1015094 | 706.8775 | 182.18344 | C6H10N6O | 1.587732534 | 0.2827 | 0.026523445↓ |
| D-Biotin | 245.0919668 | 518.395 | 244.31172 | C10H16N2O3S | 1.515098427 | 1.0857 | 0.035928770↑ |
| delta-Decalactone | 171.1378437 | 609.401 | 170.2487 | C10H18O2 | 1.849847045 | 1.5177 | 0.007156206↑ |
| D-Glucuronic acid | 193.0345141 | 562.589 | 194.1394 | C6H10O7 | 2.241280415 | 1.2756 | 0.000512579↑ |
| Dihydro-3-coumaric acid | 165.0546401 | 410.747 | 166.1739 | C9H10O3 | 1.528214666 | 1.394 | 0.037370134↑ |
| Dodecanedioic acid | 229.1439549 | 580.91 | 230.30068 | C12H22O4 | 1.66119894 | -0.9546 | 0.021378107↓ |
| Dopamine | 154.0861938 | 172.7845 | 153.1784 | C8H11NO2 | 1.825498392 | 0.753 | 0.008218865↓ |
| Formononetin | 267.066017 | 806.608 | 268.2641 | C16H12O4 | 2.025071834 | -1.1042 | 0.002867682↓ |
| Genkwanin | 283.0608516 | 871.0265 | 284.2635 | C16H12O5 | 2.062070851 | -1.1857 | 0.002214871↓ |
| Gentisic acid | 153.0181962 | 231.027 | 154.12014 | C7H6O4 | 1.835909917 | 1.2406 | 0.009038573↑ |
| Glutaric acid | 131.0337596 | 93.248 | 132.11462 | C5H8O4 | 1.55606035 | 0.6234 | 0.033445496↓ |
| Guanidinoacetic acid | 118.0613176 | 92.6126 | 117.107 | C3H7N3O2 | 1.705152566 | 1.0027 | 0.015441198↑ |
| Guanine | 150.0409746 | 176.319 | 151.126 | C5H5N5O | 2.476437968 | -1.3203 | 3.25E-05↓ |
| Guanosine | 284.0987672 | 329.596 | 283.24092 | C10H13N5O5 | 1.53671651 | 1.3507 | 0.032900583↑ |
| Hexanoylcarnitine | 260.1854318 | 582.32 | 259.1783583 | C13H25NO4 | 1.466692106 | 0.5425 | 0.043469353↓ |
| Homovanillic acid | 181.0496668 | 326.991 | 182.1733 | C9H10O4 | 1.602561332 | 0.5841 | 0.027598265↓ |
| Inosine | 267.0731584 | 331.1715 | 268.2261 | C10H12N4O5 | 1.528975498 | 2.1558 | 0.037258488↑ |
| L-Glutamic acid | 148.0602887 | 91.7513 | 147.1293 | C5H9NO4 | 1.777690879 | -1.1167 | 0.010667276↓ |
| L-Histidine | 156.0766725 | 105.9695 | 155.15468 | C6H9N3O2 | 1.603838417 | 0.5723 | 0.024721392↓ |

*(Continued)*

**Table 1.** (Continued)

| chemical compound | m/z | rt | exact mass | chemical formula | Model vs Control_VIP | log2 (FC) | p value |
|---|---|---|---|---|---|---|---|
| L-Phenylalanyl-L-Proline | 263.1388345 | 500.41 | 262.3043 | C14H18N2O3 | 2.077282892 | 0.7931 | 0.001566615↓ |
| Marmesin acetate | 289.1019761 | 693.8425 | 288.0997736 | C16H16O5 | 1.720421852 | 0.9461 | 0.014319362↓ |
| N(2)-Acetyl-L-Lysine | 189.1232505 | 106.081 | 188.22432 | C8H16N2O3 | 2.049816081 | 0.8419 | 0.001930407↓ |
| N,N-Diethyl-M-Toluamide | 192.1381451 | 1046.76 | 191.2695 | C12H17NO | 1.658473782 | 0.6825 | 0.019299997↓ |
| N-Acetyl-beta-Alaninate | 132.065568 | 184.861 | 130.1219 | C5H9NO3 | 1.573650573 | 0.5139 | 0.028179966↓ |
| N-Acetylcadaverine | 145.1334792 | 149.971 | 144.215 | C7H16N2O | 1.640234776 | 1.0602 | 0.020995681↑ |
| N-Acetyl-Glutamic acid | 190.0709275 | 195.285 | 189.1659 | C7H11NO5 | 1.606367357 | 0.6886 | 0.024447196↓ |
| N-Acetyl-L-Histidine | 198.0872059 | 106.068 | 197.19136 | C8H11N3O3 | 1.573531368 | 0.4979 | 0.028194318↓ |
| N-epsilon-Acetyl-L-lysine | 187.1078578 | 100.753 | 188.22432 | C8H16N2O3 | 2.024611789 | 0.6354 | 0.00287668↓ |
| Nicotinic acid | 124.0394187 | 147.588 | 123.10944 | C6H5NO2 | 2.168087639 | -1.2801 | 0.000737259↓ |
| N-Lactoyl-phenylalanine | 238.1071057 | 597.55 | 237.100108 | C12H15NO4 | 1.57132921 | 0.907 | 0.028460454↓ |
| Nonic acid | 187.0966503 | 487.9895 | 188.104859 | C9H16O4 | 2.208499912 | 0.5276 | 0.000690754↓ |
| O-Toluic acid | 137.059717 | 171.166 | 136.14792 | C8H8O2 | 1.805758215 | 0.7836 | 0.009169037↓ |
| Phenacylamine | 136.0745078 | 3.70776 | 135.0684139 | C8H9NO | 1.489849766 | -0.4527 | 0.039727046↓ |
| Pipecolic acid | 130.0862919 | 132.1165 | 129.157 | C6H11NO2 | 1.781759339 | 1.5181 | 0.010438908↑ |
| Quinaldic acid | 172.039368 | 306.26 | 173.1681 | C10H7NO2 | 1.669168777 | 0.7797 | 0.020624237↓ |
| Suberic acid | 173.0809113 | 144.3395 | 174.19436 | C8H14O4 | 1.612701672 | 1.3484 | 0.026434225↑ |
| Syringic acid | 197.0447419 | 256.6915 | 198.1727 | C9H10O5 | 1.538159222 | 0.7057 | 0.035930624↓ |
| Taxifolin | 303.0519806 | 562.747 | 304.2516 | C15H12O7 | 1.718717369 | 0.8785 | 0.016387198↓ |
| Threonate | 135.0287318 | 86.9696 | 136.10332 | C4H8O5 | 1.547635686 | -0.6082 | 0.034598259↓ |
| TMCA | 239.0887447 | 588.709 | 238.2366 | C12H14O5 | 1.954028191 | 1.161 | 0.003771041↑ |

glutamine levels, hypoxanthine creatinine nicotinamide in brain tissueThe content of malonic acid and dimethyl malonate increased, the content of glucodicarbonate and glycerin in liver tissues increased, and the content of glutamine hypoxanthine nucleoside asparagine asparagine and nicotinamide in kidney tissues increased. Liu Caichun Et al. [10] found that both

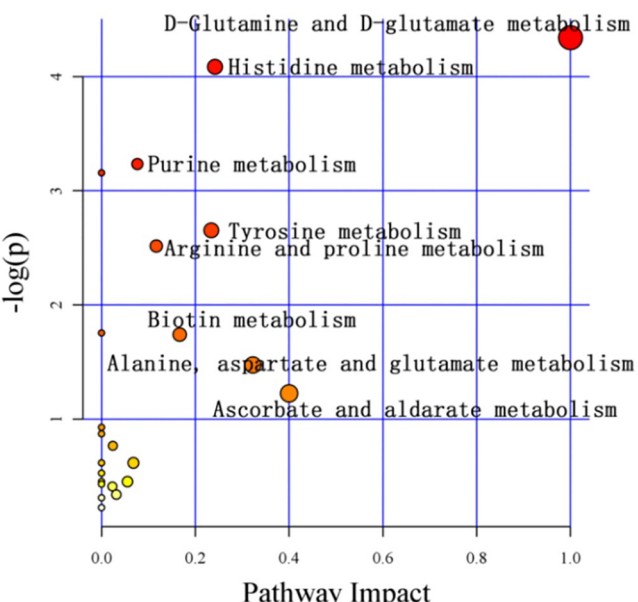

**Fig 7. Metabolic pathways of CAG rat metabolites mapped to KEGG.**

**Table 2. Differential metabolic pathways in urine of CAG rats.**

|  | D-Glutamine and D-glutamate metabolism | Histidine metabolism | Purine metabolism |
|---|---|---|---|
| p value | 0.013036 | 0.016826 | 0.039418 |
| Impact | 1 | 0.24194 | 0.076410 |
| Pathway links | http://www.kegg.jp/pathway/rno00471+C00025+C00026 | http://www.kegg.jp/pathway/rno00340+C00025+C00135+C01152 | http://www.kegg.jp/pathway/rno00230+C00020+C00212+C00294+C00242+C00387+C00147 |

electroacupuncture and moxibustion could restore various caG-induced metabolic changes, including membrane metabolism, energy metabolism and neurotransmitter function. Liu Yuetao [11] based on 1 H-NMR technical analysis astragalus chienchung soup to the CAG rats serum endogenous metabolites regulation function disorder, through multivariate statistical analysis, to clarify its regulation on chronic atrophic gastritis targets, related indicators associated with efficacy is established, by partial least-squares regression analysis and Met PA screening and treatment effect is most related metabolic pathways, find the root of remembranous milk vetch chienchung soup can obviously inhibit the CAG lesions, can obviously regulate 3—hydroxy butyric acid lactic acid acetate succinate metabolites such as disorder, the CAG treatment is the main metabolic pathways of arginine—proline metabolismGlycerol metabolism and glycine—serine—threonine metabolism pathways. Sun Yina [12] found 59 qualitative and quantitative metabolites, and then PCA OPLS-DA VIP value was used to find 8 potential differential metabolites related to dampis-heat syndrome of spleen and stomach. The metabolites of trimethyl-oxide of taurine gonosaccharide glycerol and glucose were up-regulated, and the metabolites of trimethyl-trimethyl-oxide and trigonelline phosphate creatine were down-regulated. It is concluded from the above studies that metabonomic techniques have been widely used in traditional Chinese medicine. However, 1H-NMR metabolism technology is mostly used in the studies on chronic atrophic gastritis. In this study, LC-MS technology is used to elucidate the small molecule action mechanism and related target pathways of CAG. In this study, the animal model of chronic atrophic gastritis was mainly prepared by MNNG combined with ammonia-free drinking water and hunger and satiety. The process of MNNG alkylating the DNA bases does not depend on enzymatic metabolism and can directly penetrate into the pylorus and stomach to cause canceration [13]. Alcohol can trigger acute ischemic damage to the gastric mucosa, causing damaged genes to fail to recover over time, which may be an important factor for initiating oncogenes [14]. Moreover, alcohol can accelerate the dissolution of MNNG and increase the mutation rate. Ammonia can simulate toxic damage to the stomach after *Helicobacter pylori* infection and maintain acute inflammation of the gastric mucosa [15, 16]. Ranitidine hydrochloride can inhibit gastric acid secretion, but hunger and satiety are the fusion of spleen and stomach damage. CAG is a complex disease with multiple factors and multiple genes. Compound factor modeling can simulate human disease characteristics to a greater extent and is currently the most widely used and most mature CAG model application.

Through PCA, PLS-DA and OPLS-DA LC-MS diversified analysis, using statistics, bioinformatics, chemometrics and other methods to analyze and compare the differential metabolites, the model group and the blank group of rat urine had significant metabolic differences. A total of 68 different metabolites were screened, and 23 metabolic disturbance pathways were predicted. The metabolic pathways can regulate the growth, differentiation, apoptosis and the immune system of tumor cells [17]. The statistically significant metabolic pathways are D-glutamine and D-glutamic acid metabolism, histidine metabolism, and purine metabolism. Among the metabolic pathways, the significantly different metabolites included L-glutamic

acid and 10 different products, including ketoglutaric acid, histidine, 3-methyl-L-histidine, adenosine monophosphate, adenosine, adenine, hypoxanthine, guanosine and guanine.

L-Glutamic acid, which is in the metabolic pathway of D-glutamine and D-glutamic acid, plays an important role in protein metabolism in organisms. Studies have found that L-glutamic acid can inhibit cerebral cortex, hippocampal, gastric cancer cell and neural stem cell proliferation and differentiation and induce apoptosis [18, 19]. Decreased glutamate expression levels will cause digestive system diseases. Based on this performance, L-glutamic acid is a commonly used therapeutic drug for the digestive system, especially gastric cancer and pancreatic cancer. Penicillin can induce the generation of glutamic acid and upregulate cycle-related expression genes and sugar degradation process of glucose to 2-oxoglutaric acid [20].

In the histidine metabolism pathway, 3-methyl-L-histidine, histidine, and L-glutamic acid play the role of substrate, intermediate, and product, respectively, and protein nutrition comes from the content of 3-methyl-L-histidine. Each of these compounds are effective indicators of histidine metabolic status [21]. Studies have confirmed that histidine can inhibit the proliferation and migration of lung cancer cells, thereby exerting an antitumor effect [22]. Histamine formed after the decarboxylation of histidine can relax blood vessels and is associated with inflammation. In gastritis and in the duodenum, the reaction in ulcers is sensitive. Currently, histidine is mostly used for the treatment of reducing gastric acid, relieving gastrointestinal pain and as a blood pressure treatment. L-Glutamic acid is formed after a series of processes, such as phosphoester and propionic acid formation, and its antagonists can reverse the abnormal expression of mGlu R5 and PSD-95 in the striatum of LID rats [23].

Purine metabolism provides cells with the necessary energy and cofactors to promote the growth and proliferation of cells. The most common disease with purine dysfunction is gout, and purine metabolism and its metabolites include adenosine monophosphate, adenosine, adenine, and, at times, the abnormal expression of xanthine, guanosine and guanine will promote the occurrence of gastric cancer [24]. The decomposition of purine nucleotides will promote the dephosphorylation of inosine or guanylic acid and generate inosine or guanosine, which can decompose into xanthine or guanine. The CN-II enzyme is highly expressed in tumor cells [25]. Studies have shown that purine nucleotides are essential for metabolic functions. Hypoxanthine, guanine phosphoribosyl transferase and other related purines can affect hematopoietic stem cell cycle progression, proliferation kinetics and changes in mitochondrial membrane potential [26].

## Conclusions

Metabolomics is an important technical means for studying the pathogenesis of diseases. This experiment is the first to use LC-MS metabolomics to study the pathogenesis of CAG from the perspective of urine metabolites. Fromm the method (PCA) and supervised analysis method (PLS-DA and OPLS-DA), differential metabolites of the model group and the control group were screened. These differences were mainly distributed among 23 metabolic pathways, which were glutamine metabolism with L-glutamic acid, 2-ketoglutarate in the D-glutamic acid metabolism pathway, 3-methyl-L-histidine, histidine, L-glutamic acid and purine in the histidine metabolism pathway. Adenosine monophosphate, adenosine, adenine, inosine, guanosine and guanine may be potential biomarkers for the diagnosis of CAG.

## Supporting information

**S1 Checklist.** *PLOS ONE* **humane endpoints checklist.**
(DOCX)

## Acknowledgments

The authors thank Suzhou BioNovoGene Biopharmaceutical Technology Co., Ltd., for their technical assistance. And the Editing and Manuscript Formatting service provided by American Journal Experts.

## Author Contributions

**Data curation:** Guo-Xiu Zu, Qian-Qian Sun.

**Formal analysis:** Ling Li, Hai-Liang Huang.

**Funding acquisition:** Tao Han.

**Investigation:** Liang-Kun Zhang.

**Methodology:** Guo-Xiu Zu, Qian-Qian Sun, Jian Chen, Ling Li.

**Project administration:** Xi-Jian Liu.

**Supervision:** Ke-Yun Sun, Hai-Liang Huang.

**Visualization:** Ke-Yun Sun.

**Writing – original draft:** Guo-Xiu Zu.

**Writing – review & editing:** Jian Chen, Xi-Jian Liu, Liang-Kun Zhang, Tao Han, Hai-Liang Huang.

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
