## [Decision Letter · Decision Letter 0]

26 Aug 2020

PONE-D-20-19248

Urine metabolomics of rats with chronic atrophic gastritis

PLOS ONE

Dear Dr. Tao Han,

Thank you for submitting your manuscript to PLOS ONE. After careful consideration, we feel that it has merit but does not fully meet PLOS ONE’s publication criteria as it currently stands. Therefore, we invite you to submit a revised version of the manuscript that addresses the points raised during the review process.

We look forward to receiving your revised manuscript.

Kind regards,

Tommaso Lomonaco, Ph.D

Academic Editor

PLOS ONE

Additional Editor Comments:

Dear Authors, the current version of the manuscript requires major revisions. Please address all the points raised from the reviewers. In addition, please include all the analytical figures of merit of the analytical protocol used to determine urine samples.

Best regards,

Tommaso Lomonaco

Journal Requirements:

2. To comply with PLOS ONE submissions requirements, in your Methods section, please provide additional information on the animal research and ensure you have included details on (1) methods of sacrifice, (2) methods of anesthesia and/or analgesia, and (3) efforts to alleviate suffering.

3. In your Methods section, please include a comment about the state of the animals following this research. Were they euthanized or housed for use in further research? If any animals were sacrificed by the authors, please include the method of euthanasia and describe any efforts that were undertaken to reduce animal suffering.

4. Thank you for including your ethics statement:  "Shandong University of Traditional Chinese Medicine Laboratory，Animal Ethics Committee".   

Please amend your current ethics statement to confirm that your named ethics committee specifically approved this study.

For additional information about PLOS ONE submissions requirements for ethics oversight of animal work, please refer to http://journals.plos.org/plosone/s/submission-guidelines#loc-animal-research  

5. Please upload new copies of Figures S3Fig and S6Fig as the detail is not clear. Please follow the link for more information: https://blogs.plos.org/plos/2019/06/looking-good-tips-for-creating-your-plos-figures-graphics/

Reviewers' comments:

Reviewer's Responses to Questions

**Comments to the Author**

1. Is the manuscript technically sound, and do the data support the conclusions?

Reviewer #1: Yes

Reviewer #2: Partly

2. Has the statistical analysis been performed appropriately and rigorously? 

Reviewer #1: Yes

Reviewer #2: No

3. Have the authors made all data underlying the findings in their manuscript fully available?

Reviewer #1: Yes

Reviewer #2: Yes

4. Is the manuscript presented in an intelligible fashion and written in standard English?

Reviewer #1: Yes

Reviewer #2: No

5. Review Comments to the Author

Reviewer #1: In the work under review, interesting data have been obtained on the metabolic profile of urine in chronic atrophic gastritis, which can be used for diagnostic purposes. However, I have a number of questions for the authors. For routine diagnosis of gastritis, a gastropanel is used, instrumental diagnostic methods, firstly, they pay attention to the presence of Helicobacter pylori. We would like the authors to clarify in the introduction the need for urinalysis for diagnostic purposes. What is the place of this analysis in clinical practice? Is there a real need for it? Why was the rat model chosen? Since the authors use a non-invasive diagnostic method, they could conduct appropriate research on patients, especially given that this diagnosis is widespread. What is the likelihood that the resulting metabolic profiles will match those for humans? The revealed metabolic changes can be observed with a sufficiently large number of pathologies, since they are not specific. How can these compounds be subsequently used for diagnostic purposes?

Reviewer #2: General comment:

In this work the authors studied, with the aid of LC-MS technique, urinary metabolic alterations behind chronic atrophic gastritis, a common functional gastrointestinal disorder, using rats as animal model. Since CAG is considered as a pre-cancerous state, methylnitronitrosoguanidine, a carcinogen and mutagen biochemical substance, was used to induced CAG in the animals. Following LC-MS analysis, statistical tools were applied to identify the most discriminating metabolites and the related metabolic pathways.

The experimental work was formally well conducted, and the results were sufficiently discussed. However, some issues should be solved before I could recommend publication. Also, English must be carefully revised and improved through the entire manuscript.

Specific comments:

1. PCA: The confidence ellipse in PCA graph could be useful; how have the presence of outliers been investigated?

PLS-DA and OPLS-DA: R2X, R2Y and Q2 relative to each model should be reported, as well as the number of wrong classifications in the training data set (i.e. internal validation). How many permutations have been chosen for models validation? Also, a more extensive cross-validation of the OPLS-DA model should be carried out if possible using CV-ANOVA (p < 0.01, at least) to exclude over fitting.

Independent samples T-test should be used to determine if the different biomarker candidates obtained from the OPLS-DA models are statistically significant between the two groups at the univariate level.

2. The authors should report if body mass change registered for model group was significant or not by applying a statistical test.

3. Have the LC-MS data been normalized and/or scaled? Have all features been retained, or any of them excluded if not present in most part of the samples or if their intensity was too low? Which program have the authors used for data processing?

4. Page 9 line 153: “CAG urine metabolomics analysis is corrected positive ion data”, what did the authors mean?

5. Have xenobiotics been eliminated from the list of possible metabolites? As far as I know, compounds as marmesin acetate, syringic acid o taxifolin are not endogenous. Have the authors performed potential biomarker identification with the analysis of standard compounds or MS/MS experiments? I also suggest to add to Table 1 the following information for each compound: features (tR and m/z), detected adduct, calculated m/z and delta m/z in ppm.

6. The authors could report if the metabolites belonging to the most relevant pathways were up regulated or down regulated in model group.

7. Lines 221-250 at pages 16-18 in my opinion are not useful, as they do not contribute to a critical discussion. The authors should rather discuss their results at the light of the existing literature, reporting differences respect to previous works and enlightening the novelty of their own. Several works, where authors identified potential biomarkers associated with CAG pathology in rat urine sample by NMR and LC‐MS, have not been cited nor discussed. See:

- Cui, Jiajia, et al. "NMR-based metabonomics and correlation analysis reveal potential biomarkers associated with chronic atrophic gastritis." Journal of Pharmaceutical and Biomedical Analysis 132 (2017): 77-86.

- Liu, Cai-chun, et al. "Comparative metabolomics study on therapeutic mechanism of electro-acupuncture and moxibustion on rats with chronic atrophic gastritis (CAG)." Scientific Reports 7.1 (2017): 1-11.

- Liu, YueTao, et al. "Urinary metabolomics research for Huangqi Jianzhong Tang against chronic atrophic gastritis rats based on 1H NMR and UPLC‐Q/TOF MS." Journal of Pharmacy and Pharmacology 72.5 (2020): 748-760.

- Liu, Yuetao, et al. "Material basis research for Huangqi Jianzhong Tang against chronic atrophic gastritis rats through integration of urinary metabonomics and SystemsDock." Journal of ethnopharmacology 223 (2018): 1-9.

8. Images resolution is not sufficient and should be improved.

6. PLOS authors have the option to publish the peer review history of their article (what does this mean?). If published, this will include your full peer review and any attached files.

Reviewer #1: No

Reviewer #2: No

---

## [Author Response · Author response to Decision Letter 0]

22 Sep 2020

Additional Editor Comments:Metabonomics research directions include genetic environment causes many factors, such as synthetic drugs' effects on the body of the system response, the research focus of metabonomics is medicine application is very extensive, widely applied in modern research of traditional Chinese medicine diagnosis drug toxicity evaluation of drug research and development, etc., such as researching the mechanism of action of traditional Chinese medicine (TCM) targets of traditional Chinese medicine syndrome meridian medical reports and safety evaluation of traditional Chinese medicine research field at present most of the potential biomarkers from urine, urine is not only collect invasive, but because in the urine urine rich and need a minimum of preparationTherefore, urine has been the most studied in demonstrating metabolic differences in various common and specific diseases, and the technology is now mature and has achieved good research results.

Journal Requirements:

1.The manuscript was revised in the style of PLOS ONE.

2.Before sampling, they fasted for 24 hours, drank normal water, collected urine, followed by anaesthesia with 2% pentobarbital sodium, blood collection of spleen, stomach and liver, and subsequent death.

3.Urine was collected and then used for further research.

4.Example ethics statement

This study was carried out in strict accordance with the recommendations in the Guidelines for ethical review of experimental Animal welfare(National Standard:GB/T 35892—2018). The protocol was approved by the Ethics Review Center of Shandong University of Traditional Chinese Medicine (Protocol Number:SDUTCM20190402003). All surgery was performed under sodium pentobarbital anesthesia, and all efforts were made to minimize suffering.

5.S3Fig and S6Fig

Fig. 3. Chromatogram in Total Ion Mode. 3-A: Typical Sample BPC in Positive Ion Mode, 3-B: Typical Sample BPC in Negative Ion Mode.

Fig. 6. Heat Map of the Differential Metabolites. A: Heat Map of the Differential Metabolites in CAG Rats; 6-B: Correlation Heat Map of Differential Metabolites in CAG Rats.

Reviewers' comments:

Reviewer's Responses to Questions

Comments to the Author.

The manuscript has been edited by American Journal Experts.

Reviewer #2:Urine test for diagnostic purposes: the necessity of early gastric cancer lacking characteristic symptoms, usually are diagnosed in the late, leading to poor prognosis and the prognosis of early gastric cancer was superior to that of advanced gastric cancer, so early diagnosis and treatment is the key to improving the prognosis of gastric cancer metabonomics as a new method in the field of systems biology, has become an integral part of the new tool in cancer research, and help to increase understanding of cancer pathogenesis and drug action mechanism of endogenous metabolic changes better understanding will promote the diagnosis and treatment of cancer in recent years, based on plasma urineOrganization and metabonomics analysis of gastric juice as well as the relationship between metabolic regulation and cancer research a series of progress has been made as biological specimens of detecting cancer biomarkers, urine is being more and more attention, not only because the urine is invasive collection, but also because in urine urine rich and need a minimum of preparation. 

Rats are easier to get, and also easier to detect, because of their larger size and more pronounced symptoms.Compared to mice, rats are also economically viable.

The metabolic spectrum of rats can only provide a certain direction for clinical practice, but cannot provide complete guidance for clinical practice. These compounds can at least prove the success of the chronic atrophic gastritis model in rats, which is of certain significance for the diagnosis of diseases in rats.

Reviewer #2:

Specific comments:

1.PCA incredible figure outside the confidence interval for the abnormal samples, including format conversion in data processing before peak identification filter alignment and normalization of data pretreatment, its external data check plus or minus the total ion chromatograms of display and based on mass spectrometry of metabonomics research, in order to obtain reliable and high quality of metabolomics data, usually need to quality control, quality control (QC) .When testing using QC samples quality control theory, QC samples are the same, but in the process of sample extraction test analysis will be a system error, result in QC samples, there will be differences between the smaller the difference method of stability, the higher the better data quality, reflect on the PCA analysis diagram is the concentrated distribution of QC samples, specification data reliable QC samples gathered, good repeatability, stable system.1. In order to find biomarkers, the potential characteristic peak's relative standard deviation (RSD) in QC samples, that is, the coefficient of variation should not exceed 30%. If it does, the relevant characteristic peak should be deleted Assurance(QA) to remove the features with poor repeatability in QC samples in order to obtain a higher quality data set, which is more conducive to the detection of biomarkers in QC samples,RSD＜30%;The characteristic peak proportion can reach about 70%, indicating good data . Therefore, the data reflected in PCA confidence graph are valid data, while the samples outside the confidence interval are abnormal samples.PLS-DA and OPLS-DA: R2X, R2Y and Q2 relative to each model should be reported, as well as the number of wrong classifications in the training data set (i.e. internal validation).100 permutations have been chosen for models validation. Independent samples T-test should be used to determine in the different biomarker candidates obtained from the OPLS-DA models are statistically significant between the two groups at the univariate level.

2.In the registration of weight changes at 16 weeks, normality test was performed first, and then one-way ANOVA was used to conduct statistical test of weight between the blank group and the model group, and *P＜0.01 was found between the two groups,the difference was statistically significant.

3.LC-MS data is carried out on the basis of normalization, eliminating very few data that do not exist or have too low strength. The experiment adopts most data retained after QC, QA and normalization processing.Prior to urine metabolomics analysis, the Proteowizard software (V3.0.8789) was used to convert the obtained original data into mzXML format (XCMS input file format).Using R (v3.3.2) XCMS package is used to identify the peaks identification,peaks filtration,peaks alignment, the main parameters are bw = 5, PPM = 15, peakwidth = c (10, 20), mzwid = 0.015, mzdiff = 0.01, the method = centWave.The data matrix, including mass to charge ratio (M/Z), retention time (RT) and intensity, is obtained.In the positive ion mode 22,540 precursor molecules and the negative ion mode 18,837 precursor molecules were obtained. The data were exported to Excel for subsequent analysis. In order to make comparison of data of different orders, batch normalization of data regarding peak area was conducted. 

4.The corrected positive ion data is after data pretreatment (format conversion peak recognition, filtering alignment and normalization), the data screened out have strong repeatability and good effect for urine metabolomics analysis

5.In the analysis of the diversity of urine metabolomics, potential biomarkers were identified through MS/MS experiments. Compounds were added in Table 1, including mass to charge ratio, M/Z Retention time, RT and Exact mass

6.In the * P＜0.5 and Impact＞0;The metabolic pathways of include D-glutamine and D-glutamate metabolizing,histamine and purine metabolizing pathways,In these pathways,D-glutamine and D-glutamic acid metabolism were up-regulated,three metabolites in histamine metabolism were down-regulated,and six metabolites in purine metabolism were down-regulated.

7.In recent years, metabonomics has been widely used in modern Chinese medicine treatment of chronic gastritis.Cui.Jiajia Et al.[8]found that 3 plasma biomarkers (arginine,succinate and 3-hydroxybutyrate) and 2 urine biomarkers (α-ketoglutarate and valine) might be markers of CAG in the study on plasma and urine metabolites.Chen jiaolong [9] by using nuclear magnetic metabonomics technology including observation of the stomach meridian the CAG treatment, use of electroacupuncture in the rat stomach meridians beam door and foot three mile, found that serum ghrelin level stomach metabolism of liver kidney brain cortex spectral change, the rat gastric mucosa arrangement and the thickness of the gastric mucosa has different degrees of improvement, ghrelin and substance P expression in serum increased to normal level, serum glucose glycogen content increased, the stomach tissue of glutathione and glutamine levels, hypoxanthine creatinine nicotinamide in brain tissueThe content of malonic acid and dimethyl malonate increased, the content of glucodicarbonate and glycerin in liver tissues increased, and the content of glutamine hypoxanthine nucleoside asparagine asparagine and nicotinamide in kidney tissues increased.Liu Caichun Et al.[10]found that both electroacupuncture and moxibustion could restore various caG-induced metabolic changes, including membrane metabolism, energy metabolism and neurotransmitter function.Liu Yuetao [11] based on 1 H-NMR technical analysis astragalus chienchung soup to the CAG rats serum endogenous metabolites regulation function disorder, through multivariate statistical analysis, to clarify its regulation on chronic atrophic gastritis targets, related indicators associated with efficacy is established, by partial least-squares regression analysis and Met PA screening and treatment effect is most related metabolic pathways, find the root of remembranous milk vetch chienchung soup can obviously inhibit the CAG lesions, can obviously regulate 3 - hydroxy butyric acid lactic acid acetate succinate metabolites such as disorder, the CAG treatment is the main metabolic pathways of arginine - proline metabolismGlycerol metabolism and glycine - serine - threonine metabolism pathways. Sun Yina [12] found 59 qualitative and quantitative metabolites, and then PCA OPLS-DA VIP value was used to find 8 potential differential metabolites related to dampis-heat syndrome of spleen and stomach. The metabolites of trimethyl-oxide of taurine gonosaccharide glycerol and glucose were up-regulated, and the metabolites of trimethyl-trimethyl-oxide and trigonelline phosphate creatine were down-regulated.It is concluded from the above studies that metabonomic techniques have been widely used in traditional Chinese medicine. However, 1H-NMR metabolism technology is mostly used in the studies on chronic atrophic gastritis. In this study, LC-MS technology is used to elucidate the small molecule action mechanism and related target pathways of CAG.

8.Some of the images resolution has been improved.

---

## [Decision Letter · Decision Letter 1]

14 Oct 2020

PONE-D-20-19248R1

Urine metabolomics of rats with chronic atrophic gastritis

PLOS ONE

Dear Dr. Tao Han,

Thank you for submitting your manuscript to PLOS ONE. After careful consideration, we feel that it has merit but does not fully meet PLOS ONE’s publication criteria as it currently stands. Therefore, we invite you to submit a revised version of the manuscript that addresses the points raised during the review process.

We look forward to receiving your revised manuscript.

Kind regards,

Tommaso Lomonaco, Ph.D

Academic Editor

PLOS ONE

Additional Editor Comments:

Dear Authors, please answer each question raised by the reviewer.

Regards,

Tommaso Lomonaco

Reviewers' comments:

Reviewer's Responses to Questions

**Comments to the Author**

1. If the authors have adequately addressed your comments raised in a previous round of review and you feel that this manuscript is now acceptable for publication, you may indicate that here to bypass the “Comments to the Author” section, enter your conflict of interest statement in the “Confidential to Editor” section, and submit your "Accept" recommendation.

Reviewer #1: (No Response)

Reviewer #2: (No Response)

2. Is the manuscript technically sound, and do the data support the conclusions?

Reviewer #1: Yes

Reviewer #2: Yes

3. Has the statistical analysis been performed appropriately and rigorously? 

Reviewer #1: Yes

Reviewer #2: Yes

4. Have the authors made all data underlying the findings in their manuscript fully available?

Reviewer #1: Yes

Reviewer #2: No

5. Is the manuscript presented in an intelligible fashion and written in standard English?

Reviewer #1: Yes

Reviewer #2: Yes

6. Review Comments to the Author

Reviewer #1: The article is well structured, easy to read and will be interesting to readers. The authors significantly corrected the article in accordance with the comments of the reviewers at the previous stage of the review. I believe that the article in its present form can be recommended for publication.

Reviewer #2: The authors have sufficiently addressed part of the previous comments. English has been improved, as well as the discussion. See the appended file for minor comments. Please answer to each question separately, point-to-point, and add any relevant information to the manuscript.

7. PLOS authors have the option to publish the peer review history of their article (what does this mean?). If published, this will include your full peer review and any attached files.

Reviewer #1: No

Reviewer #2: No

---

## [Author Response · Author response to Decision Letter 1]

19 Oct 2020

Response to Reviewers

· Which software have you used for statistical analysis (e.g. SIMCA)?

RE：The analysis software used for multidimensional statistical analysis is SIMCA-P (V13.0).In addition, the calculation of P value is t-test

· Univariate analysis should be conducted with p values adjusted for multiple testing.

RE：The t-test that we use for univariate statistical analysis, in general,t-test is p＜0.05 is significant, p＜0.01 is very significant, biological statistical methods are basically such a display of differences;This section provides references for significance of biological repeated screening p values.

· Please add the information about permutation test in the text. 

RE：Permutation is the result of 200 permutation tests，PLD-DA permutation test in positive ion mode is R2=（0.0，0.91），Q2=（0.0，0）PLD-DA permutation test in negative ion mode is R2=（0.0，0.9），Q2=（0.0，-0.39）

· Which signal has been selected for peak normalization? Or do the authors mean that the signals have been normalized to the sum of peak areas?

RE：According to the results, the original peak area, which is the relative strength value, is calculated，then the original peak area is standardized,batch normalization of peak area of data.Data analysis is based on standardized data.

·Have been peak areas centered and/or scaled and/or transformed before multivariate analysis?

RE：According to the results, the original peak area, which is the relative strength value, is calculated，then the original peak area is standardized,batch normalization of peak area of data.Data analysis is based on standardized data.Multidimensional statistical analysis was performed based on the data after standardized processing.

·Have xenobiotics been eliminated from the list of possible metabolites? As far as I know, compounds as marmesin acetate, syringic acid o taxifolin are not endogenous.

RE：Metabolite characterization was based on mass - MZ, RT and comparison of secondary fragments in the database.In the process of mass spectrometry comparison to the database, the exogenous substances may be characterized as the secondary debris with a higher grading value and a consistent retention time of mass and charge ratio, but it may also be other substances, which means that the mass and charge ratio and retention time can correspond to many substances. Therefore, we chose the secondary debris with a high matching degree.Therefore, there will be exogenous substances proposed by the reviewers.

---

## [Editor Report · Decision Letter 2]

27 Oct 2020

Urine metabolomics of rats with chronic atrophic gastritis

PONE-D-20-19248R2

Dear Dr. Tao Han,

We’re pleased to inform you that your manuscript has been judged scientifically suitable for publication and will be formally accepted for publication once it meets all outstanding technical requirements.

Kind regards,

Tommaso Lomonaco, Ph.D

Academic Editor

PLOS ONE

Additional Editor Comments:

Dear Authors, the current version of the manuscript is improved and thus it's now suitable to be published in PloSone Journal.

---

## [Editor Report · Acceptance letter]

29 Oct 2020

PONE-D-20-19248R2 

Urine metabolomics of rats with chronic atrophic gastritis 

Dear Dr. Han:

I'm pleased to inform you that your manuscript has been deemed suitable for publication in PLOS ONE. Congratulations! Your manuscript is now with our production department. 

Kind regards, 

on behalf of

Dr. Tommaso Lomonaco 

Academic Editor

PLOS ONE